# Geographical Range Extension of the Spotfin burrfish, *Chilomycterus reticulatus* (L. 1758), in the Canary Islands: A Response to Ocean Warming?

Fernando Espino [1],*, Fernando Tuya [1], Armando del Rosario [2], Néstor E. Bosch [3], Josep Coca [4], Antonio J. González-Ramos [4], Francisco del Rosario [2], Francisco J. Otero-Ferrer [1], Ángel C. Moreno [5] and Ricardo Haroun [1]

[1] Biodiversity and Conservation Research Group, IU-ECOAQUA, Universidad de Las Palmas de Gran Canaria, Crta. Taliarte s/n, 35214 Telde, Canary Islands, Spain; ftuya@yahoo.es (F.T.); francesco_25@hotmail.com (F.J.O.-F.); ricardo.haroun@ulpgc.es (R.H.)

[2] The Ocean Brothers, C/ Charaviscales 4, 38900 Valverde, El Hierro Island, Canary Islands, Spain; armandobrother@hotmail.com (A.d.R.); metregal@hotmail.com (F.d.R.)

[3] School of Biological Sciences, University of Western Australia Ocean Institute, Crawley (Perth), WA 6009, Australia; pol_pierce34@hotmail.com

[4] Departamento de Biología, Facultad de Ciencias del Mar, Universidad de Las Palmas de Gran Canaria, 35017 Las Palmas de Gran Canaria, Canary Islands, Spain; jcoca@pesca.gi.ulpgc.es (J.C.); antonio.ramos@ulpgc.es (A.J.G.-R.)

[5] Servicio de Impacto Ambiental, Dirección General de Lucha Contra el Cambio Climático y Medio Ambiente, Viceconsejería de Lucha Contra el Cambio Climático, Consejería de Transición Ecológica, Lucha Contra el Cambio Climático yPlanificación Territorial, C/ Profesor Agustín Millares Carló 18, 35071 Las Palmas de Gran Canaria, Canary Islands, Spain; amormar1@gobiernodecanarias.org

* Correspondence: fesprod@gobiernodecanarias.org

**Abstract:** In recent decades, numerous marine species have changed their distribution ranges due to ocean warming. The Spotfin burrfish, *Chilomycterus reticulatus*, is a reef fish with a global distribution along tropical, subtropical and warm-temperate areas of the Pacific, Indian and Atlantic oceans. In this work, we analyzed the presence of this species, between 1990 and 2019, at two islands of the Canarian Archipelago under varying oceanographic conditions: El Hierro (the westernmost island, under more tropical conditions) and Gran Canaria (a central-east island, under more cooler conditions). We expected that, under increased ocean temperatures in recent decades, the number of sightings has increased in Gran Canaria relative to El Hierro. We compiled information from different sources, including interviews and local citizenship databases. A total of 534 sightings were reported: 38.58% from El Hierro and 61.43% from Gran Canaria. The number of sightings on Gran Canaria has significantly increased through time, at a rate of 0.1 sightings per year; at El Hierro, however, the number of sightings has not significantly changed over time. Sea Surface Temperature has linearly increased in both El Hierro and Gran Canaria islands over the last three decades. Positive Sea Surface Temperature anomalies, particularly in 1998 and 2010, including high winter minimum temperatures, provide an ideal oceanographic context to favour the arrival of new individuals and, consequently, the increase in the number of sightings in Gran Canaria. Still, potential donor areas of fish recruits remain unknown.

**Keywords:** tropicalization; sea temperature rise; distribution shift; population increase; Diodontidae; Canary Islands

## 1. Introduction

One of the main effects of climate change is the progressive warming of the oceans of the planet [1]. This warming has resulted in an increase in average global sea surface temperatures, since the beginning of the twentieth century, of around 1 °C (0.89 °C in the period 1901–2012), with relevant implications for the distribution and ecology of marine species worldwide [2]. In general, the warming of the oceans, in conjunction with ocean acidification, is expected to simplify marine food webs with a reduced margin for species' acclimation [3].

The response of marine species to ocean warming is affected by multiple factors, including both biotic (e.g., thermic tolerance, reproduction type, dispersion capacity, interspecific interactions, etc.) and abiotic (e.g., availability of habitats, patterns of ocean currents, biogeographical barriers, etc.) processes [4,5]. In recent decades, numerous marine species have changed their distribution ranges because of ocean warming [6]. For those species with large dispersion capacities, poleward range extensions, and even displacements to deeper waters, have been described [7,8]. Still, some species have moved towards the equator and even underwent longitudinal displacements, because of geographical barriers and oceanographic gradients [9–11]. For instance, distribution shifts following a longitudinal gradient, with fish species approaching and moving away from the Strait of Gibraltar, have been described, which have been attributed to climate variability at global and local scales [12]. In addition to species' displacements, an overall increase in the presence and abundance of species from warm waters has been registered in temperate areas [2]. Across the world's oceans, the distributional patterns and abundances of reef fishes have been strongly influenced by ocean warming [13,14]. In turn, a range of studies have shown "tropicalization" of the fish faunas from subtropical and temperate regions of the world, including an increase in the number of species of warm-water affinities; for example, the northeastern Atlantic [15], the Mediterranean Sea [16,17], the Gulf of Mexico [18] and eastern and western Australia [19–21].

The Spotfin burrfish, *Chilomycterus reticulatus* (Linnaeus, 1758) (Tetraodontiformes, Diodontidae), is a marine fish with a circumglobal distribution in tropical, subtropical and warm temperate areas of the Pacific, Indian and Atlantic oceans [22]. In the western Atlantic, it is known from North Carolina (US) and the Gulf of Mexico, from the Florida Keys, Alabama to Corpus Christi, Texas and Campeche (Mexico). In the Caribbean, the species is cited at Panama and Bonaire and is also present off northeastern Brazil [22], including Trindade Island [23]. In the eastern Atlantic, it is known from southern Portugal, and from Cape Blanc (Mauritania) to Angola, and perhaps Namibia [24]. The species is also present in several eastern Atlantic archipelagos, including the Azores [25], Madeira [26], Selvagens [27], the Canaries [28], Cape Verde [29], Saô Tomè and Príncipe [30], and the central Atlantic islands of Saint Helena [31] and Ascension [32]. The species is considered rare in the Mediterranean Sea [33]. Adult specimens are epibenthic, reaching a maximum standard total length of about 75 cm (more common between 30 and 40 cm of total length) [34]. Individuals are found in a range of habitats, including rocky and coral reefs, at depths between 20 and 100 m, usually above 25 m, but it may occur deeper in the tropics [35,36]. The preferred temperature range for *C. reticulatus* is between 18.4 °C and 28.3 °C, with an ideal mean of 27 °C [37]. This fish feeds on hard-shelled invertebrates [33], including large-sized sea urchins [38]. Eggs and juveniles are pelagic, drifting in surface oceanic waters to about 20 cm of standard total length [35].

Despite the species being widely distributed, it is not common and, in general, the biology and ecology of this fish is not yet well known [22]. In Spain, this is a protected species by the national and regional legislation, and is categorized as 'Vulnerable' [39,40]. Most observations have been reported from the western Canary Islands, largely in overhangs, crevices and caves of rocky reefs [38,41,42]. Though the species has no commercial interest, accidental captures are not uncommon [38].

Environmental and biological data acquisition in the ocean is costly, and many times hamper the identification and census of marine life, restricting scientific inventories and collection of different sources of data [43]. However, certain charismatic mega-faunal groups, such as marine mammals, rays and sharks, are relatively easy to identify and have a considerable interest by the local eco-tourism

industry [44]. The advent of citizen science programs has boosted the collection of large amounts of data for certain faunal groups, including reef fishes [45]. In the Canary Islands, for example, citizen science programs have promoted collection of data to describe the basic ecology of the critically endangered Angelshark, *Squatina squatina* [46]. The Spotfin burrfish in an iconic reef fish species in the Canary Islands, easy to identify relative to another species of Diodontidae (*Chilomycterus spinosus mauretanicus*, *Diodon eydouxii*, *D. holocanthus* and *D. hystrix*), which are locally rare [28]. During the 1990s, *Chilomycterus reticulatus* was considered as relatively common in the westernmost islands of the Canarian Archipelago, whereas it was rare in the easternmost islands [38,41].

In this work, we initially compiled data, mostly through interviews and public citizenship databases, to compare temporal trends in the presence of the Spotfin burrfish, *Chilomycterus reticulatus*, at two island under varying oceanographic conditions in the Canarian Archipelago: El Hierro, the westernmost island, and Gran Canaria, a central-east island, subjected to more cooler conditions. We then expected that, due to increased ocean temperatures in recent decades, the number of sightings has increased in Gran Canaria relative to El Hierro. In brief, El Hierro island has long been under conditions favorable for this species, while waters off Gran Canaria island has only recently been favorable for this species, as a result of sea-water warming trends in recent decades.

## 2. Materials and Methods

### 2.1. Study Region

The Canary Islands are an oceanic archipelago of volcanic origin located in the eastern Atlantic Ocean (27.68–29.58 °N; 18.28–14.58 °W) (Figure 1). Each island has arisen from an independent volcanic system except the easternmost islands (Fuerteventura and Lanzarote), which share the same insular platform; therefore, large depths are found between adjacent islands. Fuerteventura island is located at ca. 94 km offshore the NW African coast, while El Hierro, the westernmost island, is at ca. 450 km offshore. There are strong influences of the Canary Current and the African Upwelling System on the oceanographic patterns of the Canarian Archipelago [47]. In brief, there is an oceanographic gradient in terms of Sea Surface Temperatures (SST) and productivity, with warmer (ca. 1–2 °C, occasionally 3 °C) and more oligotrophic (ca. 145 g C m$^{-2}$ yr$^{-1}$ vs. 237 g C m$^{-2}$ yr$^{-1}$) waters in the westernmost (El Hierro) relative to the easternmost islands (Fuerteventura and Lanzarote), which are under more cooler conditions [48,49]. This large-scale spatial variability across an east to west gradient influences the composition and structure of intertidal and shallow subtidal assemblages on opposite sides of the Canary Islands [50,51].

### 2.2. Data Compilation and Statistical Analysis

In this study, we took advantage of citizen science initiatives to assess potential changes in the distribution of *C. reticulatus* across the Canary Islands in recent decades. We initially gathered data from two local public citizenship databases: 'Programa Poseidon' (University of Las Palmas de Gran Canaria; www.programaposeidon.eu) and 'Red Promar' (Government of the Canary Islands; www.redpromar.com). In both databases, citizens upload data on sightings of marine species (including the island, location, date, depth, habitat, etc.). Afterwards, these sightings are checked by local experts in marine biology, specifically by marine biologists from the Las Palmas University ('Programa Poseidon') and the Environmental Administration of the Canary Government ('Red Promar'). These databases, however, contained few data on *C. reticulatus*: 0 and 8 sightings, respectively. In addition, data of 181 sightings were kindly provided by the Davy Jones Dive Centre, which daily operates in the southeast coast of Gran Canaria, so they have their own database on sightings of iconic fish species. Finally, we formally interviewed (Supplementary material: Table S1) recreational fishers (particularly, practitioners of spearfishing), commercial and recreational divers, underwater photographers and managers and divemasters of diving centers. A total of 38 interviews were carried out, for a total of 163 sightings. For each sighting, we collected information on the date, location (island, latitude

and longitude), number of fish and estimated total length (TL, categorized as: small <30 cm TL; intermediate, between 30 and 50 cm TL; large-sized >50 cm TL), type of habitat (categorized as: caves, under ledges, rocky platforms, intertidal pools, breakwaters, shipwrecks, reef covered by black corals, and sandy substrates), depth and effort (number of hours of underwater observation for each year in which an observation was reported) in the islands of El Hierro and Gran Canaria, between 1990 and 2019. The rest of the data (182 sightings) were from our own observations during SCUBA diving operations. Typically, the visibility does not differ between these two islands, which has facilitated collection of underwater data comparing the reef fish faunas between these two islands through similar methodological approaches [46].

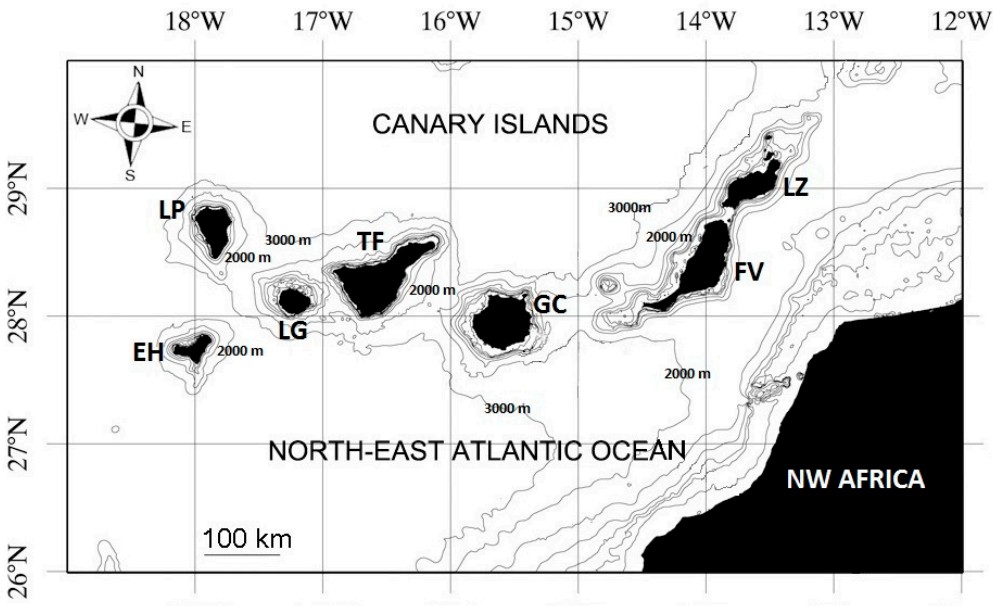

**Figure 1.** Area of study showing the location of El Hierro (EH), La Palma (LP), La Gomera (LG), Tenerife (TF), Lanzarote (LZ), Fuerteventura (FV) and Gran Canaria (GC) in the eastern Atlantic.

All sightings were standardized according to the supplied observation effort (number of hours per year); we then obtained the sightings per unit effort (SPUE), which was graphically displayed throughout the time frame of this study (1990 to 2019), separately for each island. To test for temporal trends in the number of sightings, generalized linear models (GLMs) were applied in the R vs. 3.4.3 statistical package by means of the 'R-Commander' library [52]. The sampling effort (per year) was included as a covariate. Models were separately fitted for sightings from El Hierro and Gran Canaria islands. A 'Poisson' family of errors, which is ideal for count data, with a 'log' link function, was selected. In all cases, we checked the assumptions of linearity and normality of errors through visual inspection of residuals [53]. Rather than using temperature statistics as predictor variables for each island through time, we preferred to include 'year', which encapsulates any type of oceanographic variation through time, as detailed in the next section.

## 2.3. Sea Surface Temperature and Anomalies Data

Sea Surface Temperature (SST) data (monthly L4 product code 010_001) were compiled by the marine segment of the Copernicus European system (www.marine.copernicus.eu). Monthly data files, from January 1985 to December 2018 covering each of the islands, were requested using the command line sub-setting and downloading tools. From SST monthly files, monthly SST anomalies were computed over the entire period. Then, time series for areas encompassing both El Hierro and Gran Canaria were obtained for both SST and SST anomalies.

To work out the significance of trends of these time series, we used the 'modifiemk' R library [54]. In particular, the modified Mann-Kendall method [55] was implemented to address for serial correlation

through a variance correction approach. From the Sen's slope monthly values, yearly trends were derived using the 'mmky' function. The Sen's slope provides the slope of a linear fitting to a sinusoidal curve. To analyze variation in the time series frequencies, over the covered period (1985–2018), wavelet analysis was carried out using the 'Wavelet comp' R library [56]. Wavelet analysis is based on the Fourier transformation using wavelets to provide more accurate frequency information, e.g., certain temporal patterns.

## 3. Results

### 3.1. Sightings

A total of 534 sightings were reported (Supplementary material: Tables S2 and S3), 38.58% from El Hierro (average number of hours per sighting ± SD = 13.5 ± 3.8) and 61.43% from Gran Canaria (average numbers of hour per sighting ± SD = 16.47 ± 5.1). At El Hierro, *C. reticulatus* was observed at 38 locations (Figure 2A). The mean depth of sightings was 9.9 ± 5.8 m, from a minimum of 0 m to a maximum of 30 m. Most individuals were medium (57.77%) or large (34.96%); small-sized individuals were comparatively rarer (7.29%). Only one individual (0.49%) was spotted on sandy substrates; on rocky substrates, most individuals were observed inside caves (39.33%), on rocky platforms (34.47%), under ledges (23.31%) and intertidal pools (0.98%). On breakwaters, shipwrecks and reefs covered by black corals, only one individual (0.49%) was spotted, respectively.

In Gran Canaria, the species was registered at 29 locations (Figure 2B). The mean depth of sightings was 13.1 ± 5.9 m, from a minimum of 5 m to a maximum of 50 m. Most individuals were medium (87.2%) or large (12.2%); small-sized individuals were again comparatively rarer (0.61%). Only two sightings were reported from sandy bottoms. On rocky substrates, most individuals were observed under ledges (68.9%) and inside caves (19.21%); a reduced number of individuals were spotted on rocky platforms (8.54%). A few individuals were found on breakwaters (0.61%) and shipwrecks (2.14%).

On El Hierro, the number of sightings did not significantly change through time ('Year'; estimate = $1.003 \times 10^{-2}$, $P = 0.0726$, Table 1, Figure 3A). However, the number of sightings on Gran Canaria Island significantly increased through time, at a rate of 0.1 sightings per year ('Year'; estimate = $1.004 \times 10^{-1}$, $P < 2 \times 10^{-16}$, Table 1, Figure 3B).

**Table 1.** Results of the generalized linear models (GLMs) to assess whether the number of sightings of the Spotfin burrfish, *Chilomycterus reticulatus*, changed through time ('Year') on El Hierro and Gran Canaria islands. The sampling effort through time was included as a covariate ('Effort'). Notation follows the standards of GLMs in the R statistical package.

| Site | Estimate | Std. Error | z Value | P |
|---|---|---|---|---|
| El Hierro | | | | |
| (Intercept) | $-3.497 \times 10$ | $2.015 \times 10$ | $-1.736$ | 0.0826 |
| Year | $1.800 \times 10^{-2}$ | $1.003 \times 10^{-2}$ | 1.795 | 0.0726 |
| Effort | $5.481 \times 10^{-3}$ | $3.842 \times 10^{-4}$ | 14.265 | $< 2 \times 10^{-16}$ |
| Null deviance | 247.925 | | | |
| Residual deviance | 69.953 | | | |
| Gran Canaria | | | | |
| (Intercept) | $-2.006 \times 10^{-2}$ | $2.437 \times 10$ | $-8.23$ | $< 2 \times 10^{-16}$ |
| Year | $1.004 \times 10$ | $1.213 \times 10^{-2}$ | 8.28 | $< 2 \times 10^{-16}$ |
| Effort | $2.956 \times 10^{-3}$ | $2.194 \times 10^{-4}$ | 13.47 | $< 2 \times 10^{-16}$ |
| Null deviance | 741.221 | | | |
| Residual deviance | 71.253 | | | |

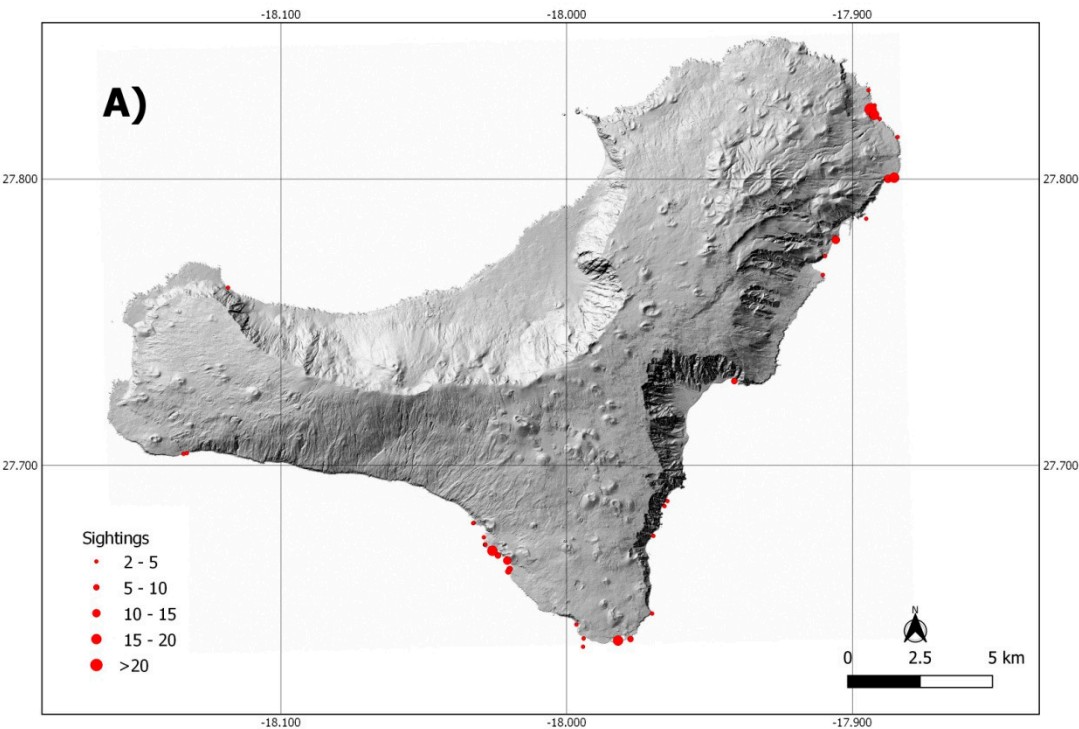

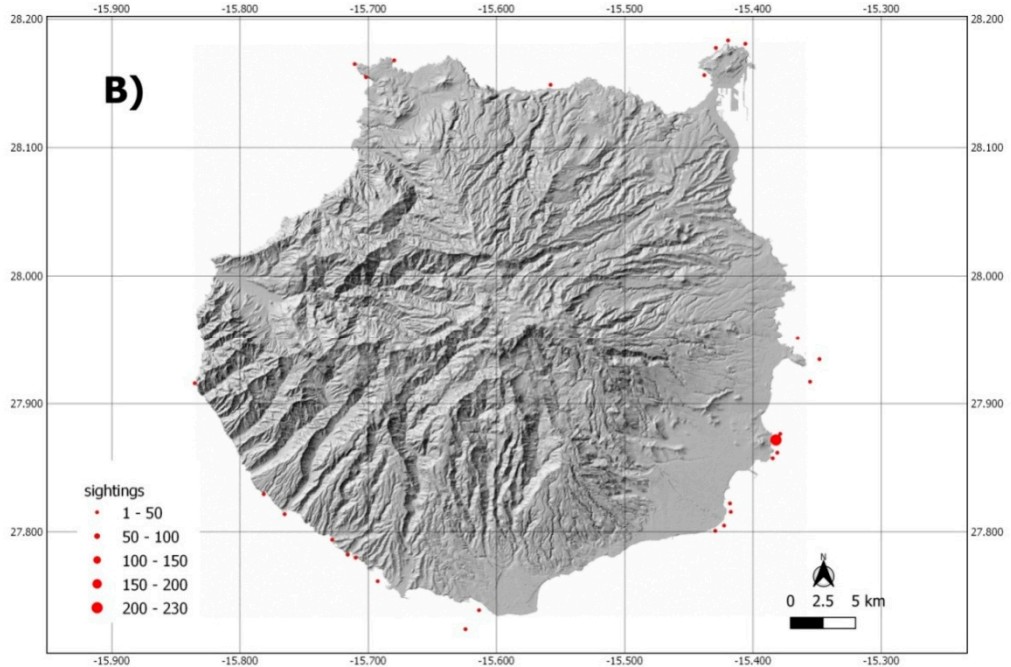

**Figure 2.** Map showing location of sightings of the Spotfin burrfish, *Chilomycterus reticulatus*, in the islands of (**A**) El Hierro and (**B**) Gran Canaria.

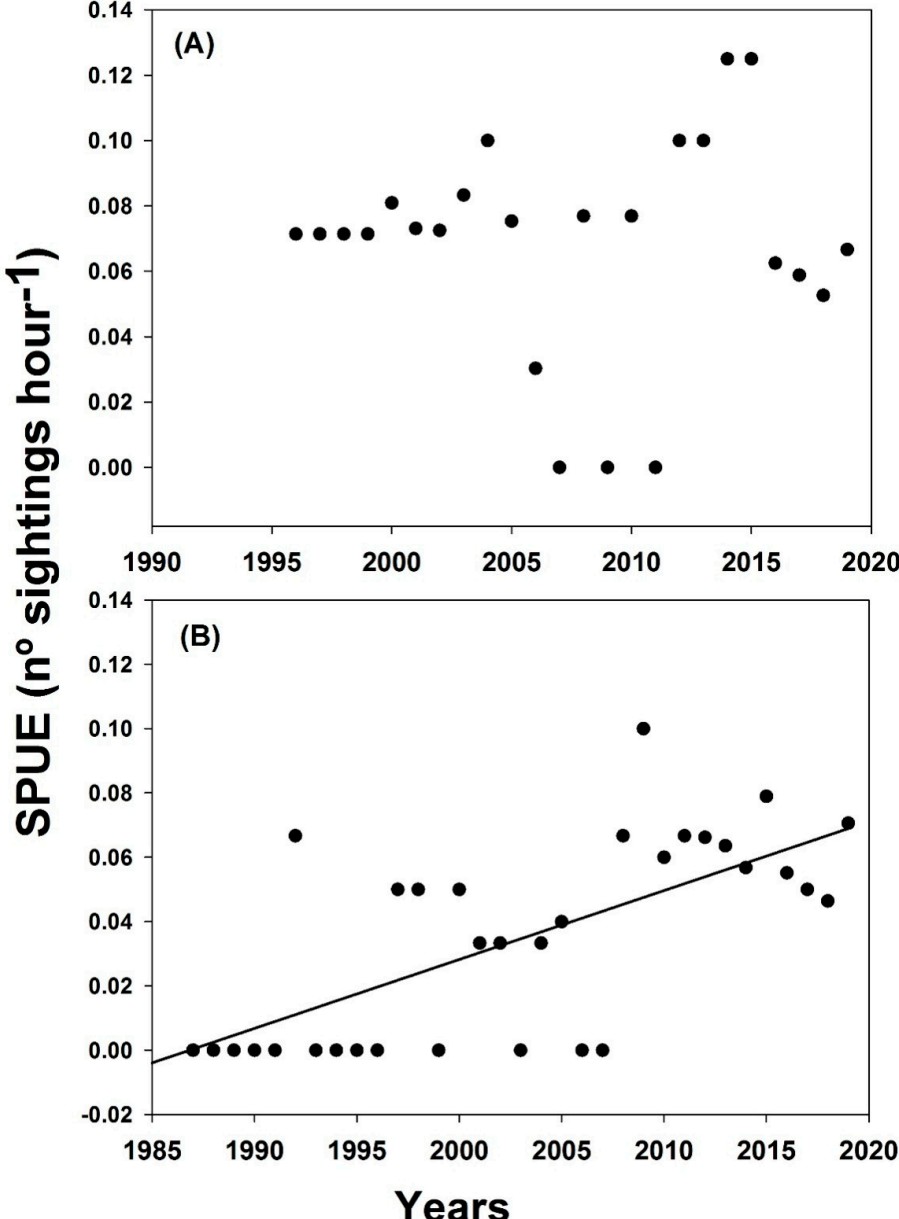

**Figure 3.** Sightings per unit of effort (SPUE), from 1990 to 2019, of the Spotfin burrfish, *Chilomycterus reticulatus*, in the islands of (**A**) El Hierro and (**B**) Gran Canaria. See Table 1 for model adjustments. The linear trend was only included for significant temporal trends.

*3.2. Patterns in Sea Surface Temperature and Anomalies*

The SST time series for both El Hierro and Gran Canaria showed a significant trend of linear increase over time (Figure 4, *p*-value < 0.01), including a yearly Sen's slope of 0.021 °C y$^{-1}$ for both islands. The SST wavelet analysis for El Hierro (Figure 5A) and Gran Canaria (Figure 5B), which accounts for significant yearly frequencies, showed a weakening for the periods around 1998 and 2010. This weakening was related to the relatively high winter temperature minima, and not high summer maxima, for these years (Figures 4 and 6). The SST anomaly wavelet analysis for both El Hierro (Figure 5C) and Gran Canaria (Figure 5D) displayed a period of intense energy around 2010, for a frequency of 18 months. In Gran Canaria, a moderate peak in energy was also observed in 1998 for a yearly frequency (Figure 5D). The high energy for these years (1998 and 2010) was connected with a relatively large period of positive SST anomalies (Figure 7).

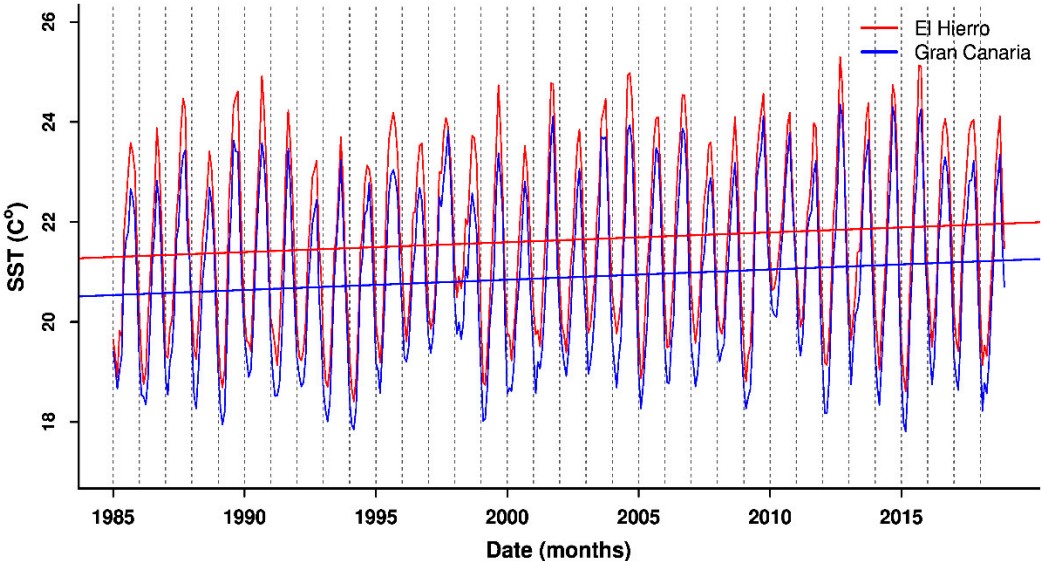

**Figure 4.** Sea Surface Temperature (SST) monthly time series for El Hierro (red) and Gran Canaria (blue), including their corresponding linear trends.

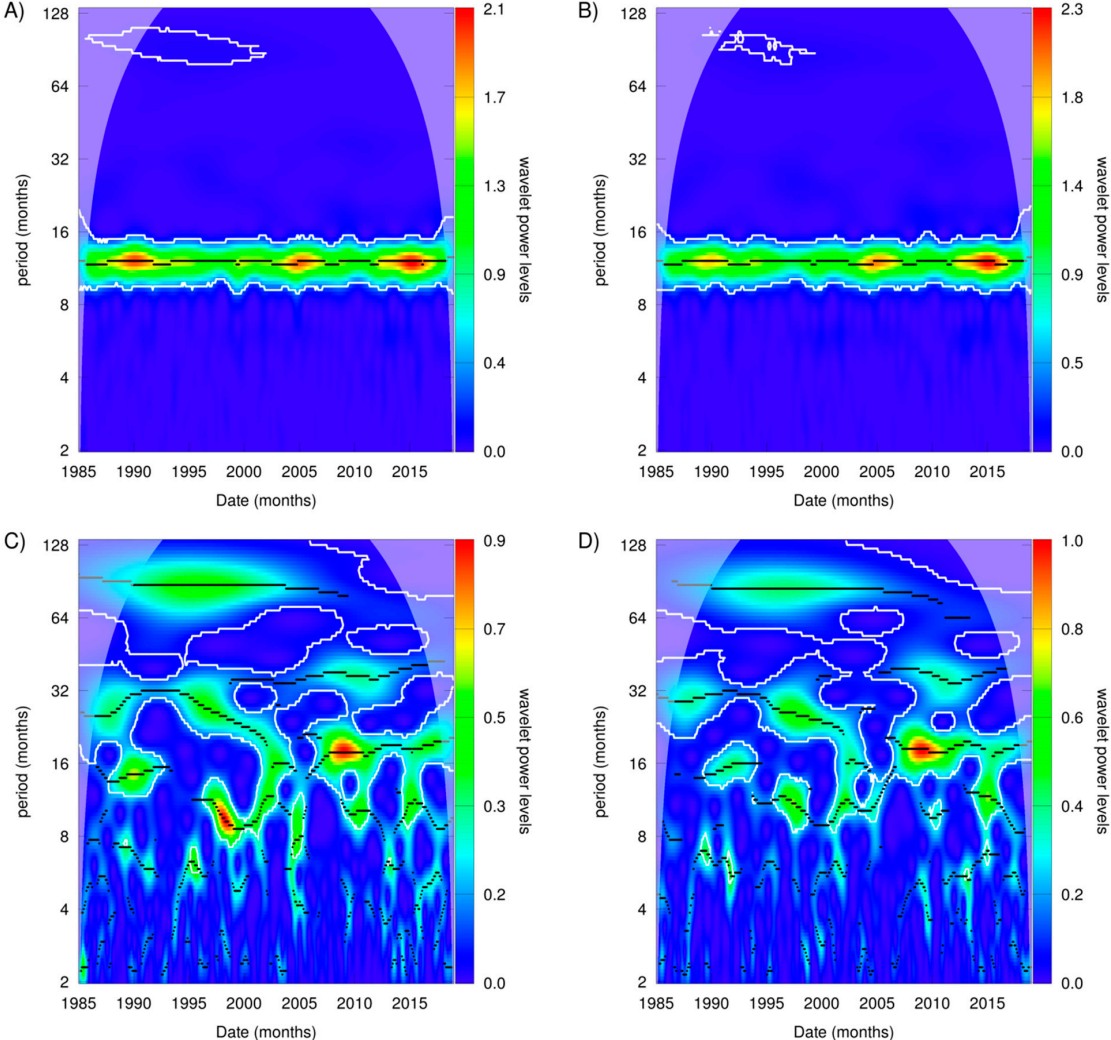

**Figure 5.** SST wavelet for (**A**) El Hierro and (**B**) Gran Canaria, and SST anomaly wavelet for (**C**) El Hierro and (**D**) Gran Canaria.

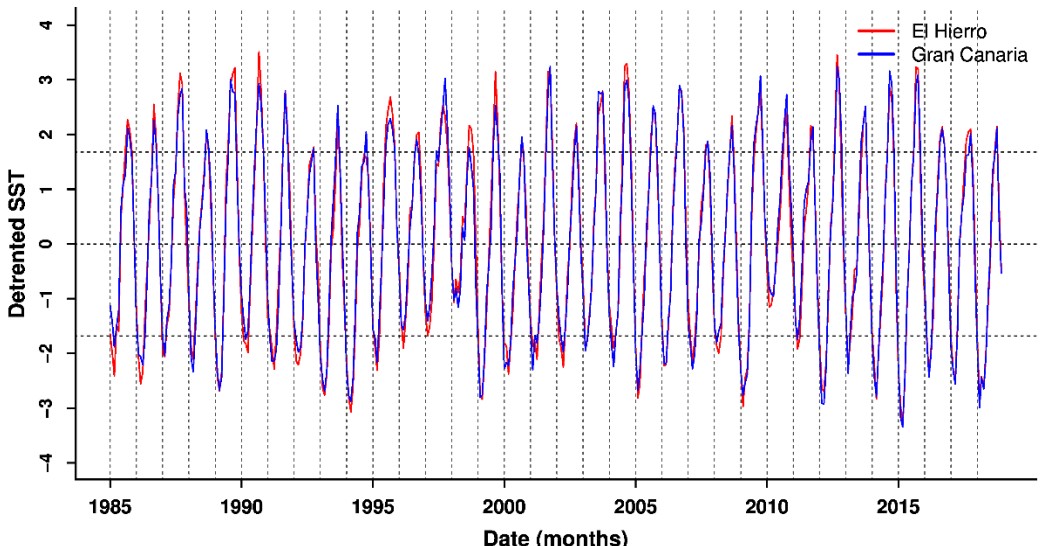

**Figure 6.** Detrended SST time series for El Hierro (red) and Gran Canaria (blue). Dashed lines accounts for +1 and −1 of the standard deviation.

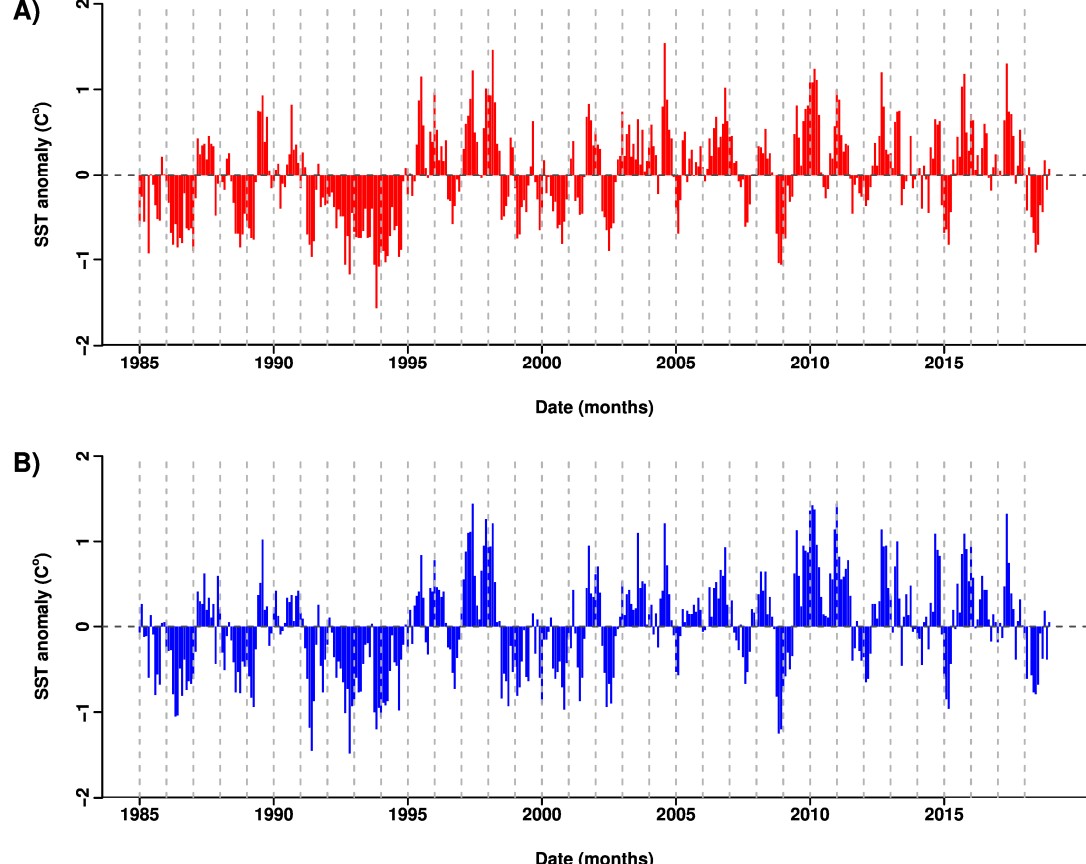

**Figure 7.** Bar-plots of SST anomalies for (**A**) El Hierro and (**B**) Gran Canaria.

## 4. Discussion

In the Canary Islands, the Spotfin burrfish, *C. reticulatus*, showed a neat spatial segregation during the 1980s and 1990s across an east to west gradient, almost restricted to the western islands, particularly at El Hierro [28,42]. In the 1980s, this fish was considered rare in the eastern islands (Fuerteventura and Lanzarote), occasional in the central islands (Gran Canaria and Tenerife) and frequent in El

Hierro [28,38]. This distribution pattern was linked with the oceanographic gradient across the Canaries, where species of warm-water affinities have been typically restricted to the western islands (e.g., *Corniger spinosus*, *Gymnothorax miliaris*), or showing increased abundances there (e.g., *Aulostomus strigosus*, *Heteropriacanthus fulgens*), including *C. reticulatus*. On the other hand, cool-water species have been typically limited to the eastern islands (e.g., *Dicentrarchus labrax*, *Sparus aurata*, *Coris julis*, *Serranus scriba*, etc.) [50,57].

Despite the lack of a conclusive theory to explain the increase in the presence of *C. reticulatus* in Gran Canaria in recent decades, increased SSTs seem to provide an ideal oceanographic context to facilitate such range expansion. The distribution of shallow-water reef fishes is considerably determined by recent patterns in SST [45]. In addition to a progressive ocean warming of the entire archipelago at a rate of ca. 0.21 °C per decade from 1985 to 2019, our results demonstrated the existence of punctual periods where warming was intense. More specifically, our data suggest that warming was particularly linked to mild winters of high temperature minima, which were particularly strong in 1998 and 2010.

On Gran Canaria, the number of sightings of *C. reticulatus* has significantly increased through the last three decades. In this island, anecdotal reports occurred until 1995. Sightings increased between 1995 and 2005, while occurrences considerably increased since 2010 to present, coinciding with periods of positive SST anomalies. This increase in SST has favored the arrival and establishment of warm-water species, particularly reef fishes [58,59]. Such a 'tropicalization' event represents both, the arrival of species of equatorial distribution ('tropicalization', in the strict sense, i.e., an increase in the ratio of tropical to temperate taxa in a given region [20]) and the geographic and population expansion of native thermophilic species ('meridionalization', as specifically used for the Mediterranean Sea). In the case of the Canaries, meridionalization involves a geographical longitudinal eastward advance of certain species, such as *C. reticulatus*, following the environmental gradient of the archipelago [58]. Other fish species have concurrently enlarged their distribution ranges eastward, such as the Goldentail moray, *Gymnothorax miliaris* [28].

At present, we have no data to unravel the origin of the *C. reticulatus* individuals observed in Gran Canaria island. It does not seem plausible that adult individuals from the western islands have moved to Gran Canaria, because adults are epibenthic and so unable to cross, at short-term scales, large (abyssal) depths between adjacent islands. However, this fish has a considerable dispersion potential because eggs and juveniles (up to 20 cm standard length) are pelagic in the oceanic surface; their distribution is therefore heavily influenced by currents [35]. As a result, increased abundances of *C. reticulatus* in Gran Canaria seems to arise from natural dispersion processes, most likely facilitated by local warming of waters.

Three potential sources of propagules may occur. Initially, the tropical Eastern Atlantic region is isolated of the warm-temperate region by the thermic front at Cape Blanco (Mauritania), which is a relevant biogeographical barrier for tropical and warm-temperate fauna [60]. However, the complex oceanographic system between the Canary Islands and the Mauritanian border [61] can ease larval dispersal northwards, facilitating the arrival, in the Canary Islands, of species of large mobility in their adult stages (e.g., *Uraspis secunda*, *Chilomycterus spinosus mauretanicus*, *Muraena melanotis*) [59]. In brief, despite the Canary Current majorly brings waters from the North, counter currents, such as the surface Mauritanian Current, which flows to the North along the NW African coast, can play a major role in tropical-subtropical water-mass exchanges [61].

A second potential source could be the tropical Western Atlantic. In addition to a large larval dispersion phase, this species has the ability to travel long distances under drifting algae [62], so it could take advantage of the Gulf Stream, the North Atlantic Current and the Canary Current to reach this archipelago. Support is provided by increased numbers of sightings for this species at the oceanic archipelagos of Azores and Madeira, both of them located in more northern latitudes, during the last decade [25,26]. This mechanism has been demonstrated for the hydrocoral *Millepora alcicornis* [63]. A less likely hypothesis points to propagules originating from the westernmost islands (e.g., El Hierro),

where the species is frequent. However, the typical current system operating in the Canarian archipelago does not seem to favor the dispersion of the species to Gran Canaria island, at least at short-term scales. Of course, molecular studies are key to shedding light on this dilemma.

The observation of *C. reticulatus* across the entire coastal perimeter of Gran Canaria island suggests that the species does not seem to have mainly arrived through maritime traffic, including oil rigs and drilling vessels, as it has been reported for some reef fishes from tropical origin near industrial ports of this island, which show larger abundances near such man-made infrastructures [64,65].

The increase in numbers of *C. reticulatus* in Gran Canaria island cannot be related to relaxed fishing efforts, because nearshore local stocks are still heavily exploited, mainly by traps [66], nor to specific conservation strategies, since there is not a specific conservation strategy for this reef fish. At present, the population structure of *C. reticulatus* at both islands seems quite similar, mostly including medium and large-sized individuals, with isolated sightings across the entire nearshore perimeter. Though there are increased abundances of the species, particularly at Gran Canaria, we here exclude the possibility of a modification in the conservation status of *C. reticulatus*. First, there is a lack of marine reserves targeting protection of this species. Secondly, this fish needs to be protected from the perspective of its role as a consumer of adults of the long-spined black sea urchin, *Diadema africanum*, a key herbivorous species, which plays an important role in determining the structure of shallow, hard-substratum, infralittoral benthic communities throughout the Canary Islands [50].

## 5. Conclusions

In this study, we have demonstrated that the distribution of the Spotfin burrfish, *Chilomycterus reticulatus*, has experienced a longitudinal displacement from the western to the central islands of the Canarian archipelago in the last three decades. Despite the biological mechanisms behind such displacement remaining elusive, an ideal oceanographic context, including large positive SST anomalies, seems to have facilitated increases in the presence of this reef fish in a central island of the archipelago.

**Supplementary Materials:** The following material is available online at http://www.mdpi.com/1424-2818/11/12/230/s1, Table S1: Questionnaire for interviews; Table S2: Data of sightings in El Hierro island; Table S3: Data of sightings in Gran Canaria island.

**Author Contributions:** F.E., F.T., N.E.B. and R.H. conceived the ideas and designed methodology; F.E., F.T., A.d.R., N.E.B., F.d.R., F.J.O.-F. and Á.C.M. collected field ecological data; J.C. and A.J.G.-R. performed, analysed and interpreted oceanographic data; Á.C.M. performed maps of sightings; F.E., F.T. and R.H. analysed sighting data and wrote the paper. All authors contributed critically to the drafts and gave final approval for publication.

**Funding:** This research received no external funding.

**Acknowledgments:** We would like to acknowledge the following people and diving centers for reporting to us their sightings: Eduardo Vera, Brian Goldthorpe, Cristina Fernández, Estanis Alemán, Tanausú Motas, Besay Ramírez, Orlando Millares, Daniel Ramírez, Álvaro Lozano, Arturo Telle, Eduardo Grandío, Enrique Faber, Jordi Lafuente, Juan Bautista Quintana, Michael Smitz, Ofelio Ranz, Pedro Almeida, Pedro Sarmiento, Ricardo Wolter, Ayose Machín, Felipe Ravina, Jaime Zamora, Miguel Maldonado and Tony Sánchez. The following diving centers are also acknowledged: Davy Jones Diving Center (Arinaga), Pandora (Arinaga) Diving Center, Blue Diving King, Buceo Norte, Canary Diving School, Extradivers Diving Center, Nautico Diving Center, Dive Academy Gran Canaria, Maldisub Productions, Pozo Scuba Diving Center, Puerto Rico Diving, Top Diving Gran Canaria, Buceo Benthos, Montagua Diving Center.

**Conflicts of Interest:** The authors declare no conflict of interest.

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
