# Peer review of "Geographical Range Extension of the Spotfin burrfish, Chilomycterus reticulatus (L. 1758), in the Canary Islands: A Response to Ocean Warming?"

_diversity, doi:10.3390/d11120230_

Round 1

Reviewer 1 Report

This paper discusses a potential geographical range extension of the spotfin burrfish from West to East in the Canary Islands.  The rationale for such a study is sound and the writing is good.  Furthermore, the general mechanisms of temperature driven range expansion are well established. However, I think the paper needs to include more detail regarding the methods- particularly more detail on how dive effort was quantified, and how potential violations of statistical independence and correlation were accounted for.  For example, trends in the dive industry, visibility, number of tourists familiar or unfamiliar with burrfish, etc. may alter the positive ID rates.  Also, the authors need to give more consideration to possible alternative hypotheses that could explain the pattern (for example, only cursory note was given to potential shifts in fishing pressure and no mention was made to changes in water visibility which could change with coastal development).  These shortcomings should be addressed before this manuscript is ready for publication. Specific comments are below.  I have made suggestions where possible regarding language changes, as there is some awkward phrasing in places.

Introduction

L62: “experimented” appears to be the wrong word here.

L97: “is widely” should be changed to “being widely”

L99: replace “in the category of” with “and is categorized as”

L100: eliminate “species”.

L101: “majorly” should be changed to “largely”.

L102: replace “Despite” with “Though”

L116: When?

L125: I am confused why a relative increase between islands is the response variable being tested.  Why not just a simple rise in sightings per unit effort at each island? Why is there an expectation that Gran Canaria will gain burrfish faster than El Hierro?

Methods

L145: Africa should probably be labeled in Figure 1 for those unfamiliar with the geography of the canary islands.  The bathymetric contour lines should also either be labeled or described.

L152: These programs should be described in more detail.  How are the data populated? Is this self reporting, or is it through a formal interview process? Are data gathered by dive operators, everyday recreational divers, etc? What kind of data are collected? What kind of quality assurance is in place?  Does visibility differ between islands, and has this visibility changed through time?

L163: Do people go on roving diver surveys or is effort only from recreational dive records?

L183: Suggest adding 1-2 sentences here about Mann-Kendall, Sen’s slope, and wavelet analysis for readers who may be unfamiliar with these methods.

Results

L 192: I cannot find Tables S1 and S2 in the supplementary materials- only copies of the figures.

L: 197: change “had either a” to “were”; change “a large size” to “large”.

L:199: Sightings data should probably be standardized for effort. How many divers dive over pure sandy habitat? I would expect most are diving on more “interesting” substrate.

L 202: Is this one individual between all three habitats? Or one individual per habitat?

L 205: change “had either a” to “were”; delete “a” before large.

L209: See comment re: line 202.

L213: An overlay of dive effort would be useful to include for this figure to see whether abundance is tracking effort spatially.

L224: Suggest adding a color legend in the corner (as you have done for Fig. 4) for those that print out a copy of your paper in black and white (it is hard to tell which color is meant to be red or blue without a legend reference).

L 238: The authors have performed a sophisticated analysis of water temperature trends and anomalies. Considering that water temperature is suggested as the main driver of increased abundance of burrfish, why is temperature (instead of year) not included as the predictor variable for the generalized linear model? 

Discussion

L 261: Do you mean Eastern instead of Western?

L 263: Suggest changing “Despite a conclusive theory” to “Though SST patterns cannot conclusively”

L 264: Replace “remains unknown, increased SSTs” with “, they”

L 266: Not sure if “conditioned” is the best word here?

L273: “In” to “on”. “anecdotical” to “anecdotal”

L274: replace “1995, in between 1995 and 2005, the number of sightings increased,” with “1995.  Sightings increased between 1995 and 2005,”

L 275: “nowadays” to “present”

L 278: replace “southern” with “equatorial” as tropicalization occurs in the southern hemisphere as well

L 279: citation needed regarding definition of tropicalization.

L 280: It is not clear how tropicalization and meridionilazation are different.  Is there a citation that introduces and defines meridonilazation?  My understanding was that meridonilization is simply tropicalization around the Mediterranean sea.

L 297: Isolated from what?

L 300: How is northward current drift possible if the Canary current brings water from the North (as described in the next paragraph)?

L 313 : Short term is repeated.

L319: Why not? Near coasts, would it not be able to disperse relatively quickly around the island, even as an epibenthic fish? These islands are relatively small.

L322: Is this relevant for burrfish, though? Is this species not poisonous? What fisheries are used on the islands? Are burrfish targeted? If not, are they a significant component of bycatch?

L323: This may be true of species-specific conservation, but species can benefit from conservation measures not targeted at them (like the establishment of fishing bans or MPAs).

L325: This may be what the data show, but these are not standardized surveys.

L327: It’s unclear what this sentence means.

Conclusions

L 338: I would soften this inference somewhat. The increased sightings do correlate with changes in SST, but almost no other variables are considered in the statistical models (like population, fishing effort, change in benthic habitat structure, etc), some of which may be correlated with temperature rises. 

Author Response

REVIEWER 1 (reviewer comments are italicized)

This paper discusses a potential geographical range extension of the spotfin burrfish from West to East in the Canary Islands. The rationale for such a study is sound and the writing is good. Furthermore, the general mechanisms of temperature driven range expansion are well established. However, I think the paper needs to include more detail regarding the methods- particularly more detail on how dive effort was quantified, and how potential violations of statistical independence and correlation were accounted for. For example, trends in the dive industry, visibility, number of tourists familiar or unfamiliar with burrfish, etc. may alter the positive ID rates. Also, the authors need to give more consideration to possible alternative hypotheses that could explain the pattern (for example, only cursory note was given to potential shifts in fishing pressure and no mention was made to changes in water visibility which could change with coastal development). These shortcomings should be addressed before this manuscript is ready for publication. Specific comments are below. I have made suggestions where possible regarding language changes, as there is some awkward phrasing in places.

Introduction

L62: “experimented” appears to be the wrong word here.

Reply. The term “experimented” was changed to “underwent”.

L97: “is widely” should be changed to “being widely”

Reply. The terms “is widely” were changed to “being widely”.

L99: replace “in the category of” with “and is categorized as”

Reply. The terms “in the category of” were changed to “and is categorized as”.

L100: eliminate “species”.

Reply. The term “species” was deleted.

L101: “majorly” should be changed to “largely”.

Reply. The term “majorly” was changed to “largely”.

L102: replace “Despite” with “Though”

Reply. The term “Despite” was replaced with “Though”

L116: When?

Reply. Since 1990, this species has been considered as rare in the eastern islands and common in the western islands (Brito, A., Falcón, J.M. Contribution to the knowledge of the distribution and ecology of Chilomycterus atringa (Pisces, Diodontidae) in the Canary Islands. Vieraea 1990, 19, 271-275) until today. We have included “During the 1990s…”

L125: I am confused why a relative increase between islands is the response variable being tested. Why not just a simple rise in sightings per unit effort at each island? Why is there an expectation that Gran Canaria will gain burrfish faster than El Hierro?

Reply. As we outline at the end of the introduction, we expect a progressive increase in the number of sightings in Gran Canaria Island in the last decades, because increased ocean temperatures in these last decades may have provided an oceanographic context that facilitate the presence of this fish at this island relative to El Hierro, which decades ago was already under more tropical conditions.

Methods

L145: Africa should probably be labeled in Figure 1 for those unfamiliar with the geography of the canary islands. The bathymetric contour lines should also either be labeled or described.

Reply. Figure 1 was changed, including now “NW AFRICA”, “2000 m” and “3000 m”, as bathymetric lines labels.

L152: These programs should be described in more detail. How are the data populated? Is this self reporting, or is it through a formal interview process? Are data gathered by dive operators, everyday recreational divers, etc? What kind of data are collected? What kind of quality assurance is in place? Does visibility differ between islands, and has this visibility changed through time?

Reply. In our study, all sighting data are included in Table S1 (El Hierro) and Table S2 (Gran Canaria). The “Programa Poseidon” is a database of the University of Las Palmas de Gran Canaria and “Programa Red Promar” is a database of the Environmental Department of the Canary Government. In both, citizens can communicate or upload data on sightings of marine species (including island, location, date, depth, and habitat, etc.). Afterwards, these data are checked and confirmed by local experts in marine biology/ecology. Nevertheless, these databases contain few data on sightings of C. reticulatus: 1 and 6, respectively. Most data on sightings (347) were here obtained through interviews with local divers, underwater photographers, instructors and dive masters of local diving centers and practitioners of spearfishing. The interview was formal and included the following questions: Island, location, date, depth, number of individuals and size (large>50 cm TL, 30 cm TL<medium<50 cm TL or small<30 cm TL), habitat (cave, under ledge, rocky platform, intertidal pools, breakwaters, shipwrecks, reefs covered by black corals, and sandy substrates, and effort (number of hours of observation for each sighting). The rest of the data (180 sightings) were from our own observations (i.e. Armando del Rosario; Francisco del Rosario, Néstor Bosch and Fernando Espino). The visibility does not differ between islands. Evidence for this is provided from a range of studies comparing the reef fish faunas of islands, including nearshore sharks through a citizen program, as indicated in reference 46. All this information has been now included.

L163: Do people go on roving diver surveys or is effort only from recreational dive records?

Reply. Most sightings were provided by recreational dive records, i.e. people diving at their own criteria and without any specific goal.

L183: Suggest adding 1-2 sentences here about Mann-Kendall, Sen’s slope, and wavelet analysis for readers who may be unfamiliar with these methods.

Reply. We have included that:

- “Wavelet analysis is based on using the Fourier transformation using wavelets to provide more accurate frequency information, e.g. certain temporal patterns.

- The Sen’s slope provides the slope of a linear fitting to a sinusoidal curve.

Results

L 192: I cannot find Tables S1 and S2 in the supplementary materials- only copies of the figures.

Reply. Tables S1 and S2 are included in an Excel file; these tables contain raw data for each sighting. This supplementary material was uploaded to the Editorial on-line platform together with the manuscript.

L: 197: change “had either a” to “were”; change “a large size” to “large”.

Reply. All these changes were included.

L:199: Sightings data should probably be standardized for effort. How many divers dive over pure sandy habitat? I would expect most are diving on more “interesting” substrate.

Reply. Effectively, in turn, we present our data (see Figure 3) as number of sightings per unit effort.

In the Canary Islands, most divers indeed dive on rocky substrates, which offer habitats of large fish diversity. In some places, however, this type of bottom is adjacent to sandy bottoms, so a few individuals of C. reticulatus were seen over sandy habitats.

L 202: Is this one individual between all three habitats? Or one individual per habitat?

Reply. One individual per habitat

L 205: change “had either a” to “were”; delete “a” before large.

Reply. All these changes were included.

L209: See comment re: line 202.

Reply. On breakwaters: 2 sightings; on shipwrecks: 7 sightings.

L213: An overlay of dive effort would be useful to include for this figure to see whether abundance is tracking effort spatially.

Reply. The potential confounding of varying sampling effort through time is, indeed, accounted in the GLMs where sampling effort (number of hours) was included as a covariate. For the sake of simplicity, we prefer to maintain the plot as it stands.

L224: Suggest adding a color legend in the corner (as you have done for Fig. 4) for those that print out a copy of your paper in black and white (it is hard to tell which color is meant to be red or blue without a legend reference).

Reply. Figure 3 was changed for a better understanding and is all in black and white.

L 238: The authors have performed a sophisticated analysis of water temperature trends and anomalies. Considering that water temperature is suggested as the main driver of increased abundance of burrfish, why is temperature (instead of year) not included as the predictor variable for the generalized linear model?

Reply. Basically, because temperature and years are correlated. We could have included temperature data in the GLMs, as indicated. However, it is difficult to decide to pick up a specific temperature data per year, e.g. mean, max, min, detrended seasonal max, detrended seasonal min, etc. In brief, we prefer to include “year” in the GLMs, which is very easy to understand, and connect what happened, in terms of temperatures anomalies, in the last decades, with observed patterns.

Discussion

L 261: Do you mean Eastern instead of Western?

Reply. There is a mistake here. "western" was changed to "eastern"

L 263: Suggest changing “Despite a conclusive theory” to “Though SST patterns cannot conclusively”

Reply. We prefer to maintain the original sentence.

L 264: Replace “remains unknown, increased SSTs” with “, they”

Reply. We prefer to maintain the original sentence.

L 266: Not sure if “conditioned” is the best word here?

Reply. Replaced by “determined”.

L273: “In” to “on”. “anecdotical” to “anecdotal”

Reply. These changes were included.

L274: replace “1995, in between 1995 and 2005, the number of sightings increased,” with “1995. Sightings increased between 1995 and 2005,”

Reply. These changes were included.

L 275: “nowadays” to “present”

Reply. “nowadays” was changed to “present”.

L 278: replace “southern” with “equatorial” as tropicalization occurs in the southern hemisphere as well

Reply. We agree with the reviewer in this point and this has been changed.

L 279: citation needed regarding definition of tropicalization.

Reply. We included a definition of tropicalization: “an increase in the ratio of tropical to temperate taxa in a given region”; reference bis provided by Wernberg et al. 2013.

L 280: It is not clear how tropicalization and meridionilazation are different. Is there a citation that introduces and defines meridionalization? My understanding was that meridionialization is simply tropicalization around the Mediterranean sea.

Reply. We now explain in the text the differences between the two processes. “Tropicalization” has been defined as an increase in the ratio of tropical to temperate taxa (here fish, in particular) in a given region (Wernberg et al. 2013). In the Canary Islands, the intrusion of “new fish species” of equatorial distribution has been recorded, see Brito, A.; Moreno-Borges, A.; Escánez, A.; Falcón, J.M.; Herrera, R. New records of Actinopterygian fishes from the Canary Islands: tropicalization as the most important driving force increasing fish diversity. Rev. Acad. Canar. Cienc. 2017, 29, 31-44.

Meridionalization” has been defined as the northern spread of the “native warm water biota” in the Mediterranean Sea (Azzurro, E. 2008. The advanced of thermophilic fishes in the Mediterranean Sea: overview and methodological questions. CIESM Workshop Monographs 35: 39-45), as the reviewer pointed. This process occurs in the Canary Islands too, but the spread of the native thermophilic fish species is from western to eastern island (see Brito et al. 2017), not to northward as in the Mediterranean.

L 297: Isolated from what?

Reply. Isolated of the warm-temperate region in the Eastern Atlantic...we have included “...isolated of the warm-temperate region….”

L 300: How is northward current drift possible if the Canary current brings water from the North (as described in the next paragraph)?

Reply. There is a complex oceanographic system between the Canary Islands and the Mauritanian border. Overall, the Canary Current brings waters from the North, as the reviewer pointed. The Cape Verde Frontal Zone separates waters from tropical (southern) and subtropical (northern) origin, but there are counter currents, as the Poleward Undercurrent and the surface Mauritanian Current along the NW African coast, which flow to the north. This plays a major role in tropical-subtropical water-mass exchange. See Peña-Izquierdo, J.; Pelegrí, J.L.; Pastor, M.V.; Castellanos, P.; Emelianov, M.; Gasser, M.; Salvador, J.; Vázquez-Domínguez, E. The continental slope current system between Cape Verde and the Canary Islands. Scientia Marina 2012, 76S1, 65-78.

L 313: Short term is repeated.

Reply. “Short term” was deleted.

L319: Why not? Near coasts, would it not be able to disperse relatively quickly around the island, even as an epibenthic fish? These islands are relatively small.

Reply. Because there are no gradients in abundance as reported for other reef fishes, which include larger abundances around man-made structures. This has now been pointed in the manuscript.

L322: Is this relevant for burrfish, though? Is this species not poisonous? What fisheries are used on the islands? Are burrfish targeted? If not, are they a significant component of bycatch?

Reply. This is relevant because C. reticulatus can be captured accidentally by some fishery gears (e.g. nets). Traditionally, some professional fishers collected this fish as an ornamental object. Actually, only accidental captures occur; the gears most employed in the islands are fish traps and trammel nets. It is not a significant component of the bycatch, but we know that some recreational spearfishes capture this fish with ornamental purposes. This species is not poisonous. To shed light on this we have included the following reference: A. García-Mederos, Tuya, F., Tuset, V.M. 2015. The structure of a nearshore fish assemblage at an oceanic island: insight from small scale fisheries through bottom traps at Gran Canary Island (Canary Islands, eastern Atlantic). Aquatic Living Resources 28:1-10.

L323: This may be true of species-specific conservation, but species can benefit from conservation measures not targeted at them (like the establishment of fishing bans or MPAs).

Reply. In Gran Canary Island, there are not marine reserves. Professional and recreational fishing is permitted all around the island, except spearfishing that is restricted only to few areas. Marine Special Areas of Conservation (SACs) are the only areas with general measures for the conservation of some habitats (i.e. seagrass habitats) or species (i.e. cetaceans and marine turtles). These measures do not affect the conservation status of C. reticulatus.

L325: This may be what the data show, but these are not standardized surveys.

Reply. As well pointed by the reviewer, data indicate that individuals are medium or large-sized, appear isolated of other individuals, and across the entire nearshore perimeter of the islands.

L327: It’s unclear what this sentence means.

Reply. Despite the increasing in the number of sightings of the burrfish in Gran Canaria Island, we do not recommend a change in the conservation status of the species. This is because, as we state in the introduction, this fish is a protected species by the national and regional legislation, and is categorized as 'Vulnerable' [39,40].

Conclusions

L 338: I would soften this inference somewhat. The increased sightings do correlate with changes in SST, but almost no other variables are considered in the statistical models (like population, fishing effort, change in benthic habitat structure, etc), some of which may be correlated with temperature rises.

Reply. We still prefer to maintain this sentence, as changes in habitat structure and fishing effort through the last decades are unlikely to occur in the area, as we pointed in other parts of the paper.

Reviewer 2 Report

Read very well. A thorough piece of work.

Author Response

REVIEWER 2

The reviewer did not provide anything to correct.

Reviewer 3 Report

Please see my review document.

Author Response

REVIEWER 3 (reviewer’s comments are italized)

Specific:

Line 104: The sentence with the phrase “as no variability cannot be observed” is quite confusing wording…

Reply. The sentence has been eliminated.

Line 215: The simple approach for handling the sightings data is in contrast to the overblown approach for SST where wavelet analysis was performed. Yes, a GLM is a fine statistical technique, but the data are not very convincing visually for this finding of trend.

Reply. We somehow emphasize with this view. We believe GLMs, in particular with a Poisson error structure, are ideal to handle our data, due to overdispersion by a high number of 0s. In any case, as we point out in the next point, we have re-plotted our sighting data through time to shed light on a better visualization of temporal patterns.

Line 224: The use of bar charts for this type of data is not ideal. For example, are there any real zeros here? By real zeros, I mean some effort existed but no fish were observed. This needs to be distinguished from situations where there was no effort. In present form, this figure does not convey to me that there has been any significant change in sightings over the years for either site, despite what the GLM says. As I read thru more of the paper, I keep coming back to this figure to look for the increase in sightings. I strongly advise redoing this figure to show any zeros that are in the survey data. And overlay trend lines from the GLM. Essentially that is a regression, correct? So, it would be possible to show the trend lines?

Reply. The reviewer is absolutely correct and so we have replaced this bar chart by a scatterplot containing all data, including the 0s. We agree with the reviewer that this new plot is much straightforward and neat to visualize temporal patterns. We have accordingly changed the legend of this new Figure 3.

To expand on the previous, to force a linear construct on this is also questionable. Populations do not generally behave in nice straight lines. There are many ways to incorporate nonlinearity and this should be investigated. A GLM may be a good start but there should be much more exploration here. Perhaps asymptotic, perhaps a breakpoint, but hard to advise further without seeing the data… The SPUE is much like a CPUE used in fishery stock assessments. Normally, the preparation and standardization of a CPUE time series takes more analysis than what is in this entire manuscript, yet we are just presented with an SPUE time series with no discussion of what goes into it, aside from a few sentences in section 2.2. These indices are never as simple as just adding up something for a year, something else for a year, and dividing the 2. At very minimum show the numerator and denominator series separately first. Do a better job of talking away any observer, water clarity, etc. bias. Also, to further this point, 3 of the early data points for Gran Canaria appear to be exactly the same value of 0.05, so I suspect this was a single sighting over some equally invariant effort denominator for 3 years in a row. Then the next 3 non-zero values were also at a lower but exact same magnitude, making the reader very suspicious of the resolving power of this index and what numbers are being used in the calculation.

Reply. We agree with the reviewer that population dynamics is far away from linearity. However, for the sake of simplicity and parsimony, linear adjustments provide straightforward results, i.e. maximum explanation with minimum model complexity. We believe this is particularly our case-study, because we just want to show a progressive increase in the presence of our target species in a particular island relative to the other island. GLMs somehow encapsulate non-linearity. In turn, we applied a range of candidate models to our data, including GAMs, but model parsimony did not increase. In turn, the decrease in deviance from the null to the fitted model is quite large (please, see results presented in Table 1). In other words, we can fit alternative models that explain more variance of our data, but at the cost of increasing the complexity of the model by adding more parameters. By presenting the new Figure 3, we believe the data can be better visualized and interpreted. The reviewer is correct in pointing out that for the 3 early observations in Gran Canaria, we just had one observation over the same effort. There is little we can do in this sense, as this old data was provided by the same observed. In any case, over and above all this issue, we still believe readers can grasp the idea we want to get across: that the number of sightings of our target fish species has increased in the last decades in the island of Gran Canaria, which is connected with a climatic scenario that seems to promote such an increase.

Line 233: The wavelet analysis does not contribute much to this other than to add some flashy color visuals. The SST anomaly wavelet has intriguing patterns but these are not developed at all in the text. It is nice that these analyses were accomplished but they don’t add anything. A seasonal decomposition analysis with an accompanying figure of 2-3 panels (seasonal cycle, yearly trend, and perhaps the residual) would be much more useful than the existing 4 panels of Figure 5. This would cleanly show the seasonal cycle plus an interannual pattern, and could be easily accomplished in R, or for example in a GAM.

Reply. We here don’t agree with the reviewer. We believe the wavelet analysis contribute to identify, in conjunction with the SST anomalies plots, those “uncommon” years in terms of sea water temperatures. We then link these years, in particular 1998 and 2010, with increases in the number of fish sightings in Gran Canaria Island. Including a seasonal decomposition analysis would not provide much more, because we are looking at long-term trends.

Line 246: If there is compelling difference in SST anomaly between the 2 sites, it is very difficult to pick up from looking at this Figure, and the reader is more inclined to think these 2 sites are identical oceanographically when that is not the point the authors are attempting to make.

Reply. Of course, the two islands are relatively close, so it is expected that overall SST anomalies are similar. For example, if a winter is mild, the winter would be mild in both islands. Hence, it is obvious that readers a priori think the oceanography of both islands is similar. However, over and above this fact, there is an overall larger SST in El Hierro Island relative to Gran Canaria, as it can be easily observed in Figure 4. In summary, such a progressive increase in SST in Gran Canaria, in particular in 1998 and 2010, seem to provide an oceanographic scenario that facilitate the presence of our target fish in the last decades.

General:

Despite the statistics in the table, I am not convinced of an increase in sightings. The authors do a poor job of conveying that information. I have serious concerns over their SPUE. The analysis on the environmental side of things is pretty good, but of no use if there is no compelling biological story.

Reply. We believe the new plot (Figure 3) do a much better job to convince readers of the increase in sightings in Gran Canaria Island.

Summary: I think that this could be a valuable contribution but I would need to be way more convinced that there is something in the SPUE data. As it stands, that portion of the manuscript is much too weak and is an essential component for this to move forward.

Reply. Please see this reviewed version and our replies to all reviewers.

Round 2

Reviewer 1 Report

The authors have made a good effort to improve the manuscript and have answered many of my previous concerns. There are still a few things that need clarification.  In general, I think the forays into intra-island dispersal and abundance patterns do not add much to the main thesis of the manuscript but instead generate more problems than they solve (see below). If these issues are addressed (or if this side part of the manuscript is reduced or eliminated) I support this manuscript be published in Diversity.

Specific comments:

L 125: The author’s response makes sense and should be incorporated into the end of the introduction explicitly.  Specifically, the authors need to make more clear that El Hierro is already under tropical conditions favorable for this species, and that Gran Canaria is just approaching that threshold.

L 166: Thank you for providing more details regarding the methods. I have a few follow up questions:

              How are data checked by local experts? What is checked for?

              How many formal interviews were completed?

              Can a copy of the questionnaire be included as a supplementary material?

              Were your own observations also standardized for effort? Were these recreational dives, dives   for other projects, or dives specifically to find this species (i.e. roving diver surveys)?

L 185: You mention that reference 46 indicates visibility does not differ between islands, but that manuscript does not appear to mention visibility at all.

L 238: I see the reviewer’s argument regarding all of the potential “mini-variables” surrounding including temperature as an effect of the model, and in using year instead to encapsulate all of that variability. Please include this explanation in the text, probably in the last paragraph of section 2.2.

L 240: I understand the concerns of the authors regarding simplicity of the figure. However, It is important that these plots speak in the same language as the figures and the models.  As it stands right now, figure 2 is not particularly useful (beyond providing an outline of the islands) because it does not show effort.  Furthermore, it does not show where the zeroes are.  For example, as someone who has never been to El Hierro, I do not know if the north end of the island has no burrfish because there are none there, or because there were no dive surveys there.  If the authors wish to maintain this spatial intra-island scale data in these figures, the data need to reflect 1) the dives with no sightings and 2) the sightings data expressed as SPUE.  One way to do this without adding much complexity would be to have the size of the circle correlate to the number of hours of diving performed at a site, and a color gradient for SPUE.  This would give the reader an idea of the spatial extent of the sampling effort as well as a true estimate of population gradients of the fish, which the authors discuss elsewhere to make some of their discussion points.  Alternatively, the authors could eliminate the references to the intra-island spatial patterns in the manuscript and focus on the between-island patterns.  This may be necessary as it appears in the supplementary data that there are no data on dives undertaken when fish were not sighted (so again, I cannot tell whether the spatial gaps in coverage were because of a lack of effort or because dives occurred but there were no fish sighted).

L 298: The grammar of this first sentence needs to be fixed.

L 316: If the authors wish to use Meridionalization I think it is important to describe it is a Mediterranean specific term (with the 2008 reference) for readers who are not from the Mediterranean region and may not be familiar with it.  The authors’ response clarified the distinction and it would be good to clarify the text as well.

L342: “of” to “from”

L348: replace “despite” with “though”

L365: I disagree with the authors that the lack of accumulation of this fish around man-made structures or abundance gradients are evidence that it was not introduced by maritime traffic, as transport and aggregation are two separate processes.  Given the long larval dispersal time of this species, is it not possible that untreated ballast water could play a role in larval (or even juvenile) dispersal?  Furthermore, if the fish are not aggregating, doesn’t this make the point that they are able to freely disperse? I suggest removing this paragraph or including an analysis with “distance to port” included as a co-variate if the authors wish to make this strong of a case.  Without data on effort across the island ( including where the fish are not seen) it is unclear how effort differs spatially.  Perhaps the patterns really are that clear; but right now they don’t come through to a reader of the manuscript.  I think revising figure 2 as requested above would be a good way to convince readers of the patterns the authors describe.

L 372: Please also mention the lack of marine reserves in the text as you did in your response to the reviewer, since this is probably the main way that these fish would piggyback off of non-specific protections.

Supplementary material: I am a bit confused about the format of the dataset and would appreciate some clarification.  Each row shows exactly one and only one burrfish sighted despite a wide range in effort. Were there never more than 1 burrfish sighted per operator per year, or per dive?  Furthermore, what about locations where diving occurred but burrfish were never spotted?

Author Response

REVIEWER 1

The authors have made a good effort to improve the manuscript and have answered many of my previous concerns. There are still a few things that need clarification. In general, I think the forays into intra-island dispersal and abundance patterns do not add much to the main thesis of the manuscript but instead generate more problems than they solve (see below). If these issues are addressed (or if this side part of the manuscript is reduced or eliminated) I support this manuscript be published in Diversity.

Specific comments:

L 125: The author’s response makes sense and should be incorporated into the end of the introduction explicitly. Specifically, the authors need to make more clear that El Hierro is already under tropical conditions favourable for this species, and that Gran Canaria is just approaching that threshold.

Reply. We have accordingly included, at the end of the introduction, the following sentence: In brief, El Hierro island has long been under conditions favorable for this species, while waters off Gran Canaria island has only recently been favorable for this species, as a result of sea-water warming trends in the last decades”.

L 166: Thank you for providing more details regarding the methods. I have a few follow up questions: How are data checked by local experts? What is checked for?

Reply. Any citizens can report sightings of marine species in these databases; when a sighting is reported, the experts responsible of the databases (marine biologists) of the Las Palmas University (PROGRAMA POSEIDON) or the Environmental Administration of the Canary Government (RED PROMAR) contact via email with the citizen and request data on sightings (e.g. island, location, date depth, habitat, behaviour, number of dives and time, etc. Basically, they check the validity of each sightings based on (i) photographs taken by the observes and (ii) any other trait/description of the animals. In any case, data on sightings of the burrfish in these databases were very scarce, as the reviewer can see in the Supplementary material (Tables S1 and S2; 0 and 8 sightings, respectively). This has been clarified in the methods.

How many formal interviews were completed?

Reply. Of the 534 sightings, 182 were from our own observations; 181 sightings were kindly provided by Davy Jones Diving Centre (this dive centre operates every day in the southeast coast of Gran Canaria Island), so they have their own database; 8 sightings came from Red Promar, and the rest of sightings (163) comes from 38 interviews. To shed light on this, we included these data in material and methods (lines 160 to 173).

Can a copy of the questionnaire be included as a supplementary material?

Reply. We have included the questionnaire as a Supplementary material: Table S1. Tables S1 and S2 were changed to Tables S2 (El Hierro) and S3 (Gran Canaria).

Were your own observations also standardized for effort? Were these recreational dives, dives for other projects, or dives specifically to find this species (i.e. roving diver surveys)?

Reply. Yes, our own observations were standardized according to the number of diving hours; these dives were for other projects, not specifically to find this species.

L 185: You mention that reference 46 indicates visibility does not differ between islands, but that manuscript does not appear to mention visibility at all.

Reply. There are many studies that successfully applied underwater visual census for fishes in the Canary Islands, particularly in El Hierro and Gran Canaria Islands (e.g. Bortone et al. 1991. Scientia Marina 55(3): 529-541; Bortone et al. 1994. Bulletin of Marine Science 55(2-3): 602-608; Falcón et al 1996. Marine Biology 125(2): 215-231). We just included reference 46 as an example. As the reviewer knows, this sampling method requires clear waters, so we think that reference 46 is enough in this sense.

L 238: I see the reviewer’s argument regarding all of the potential “mini-variables” surrounding including temperature as an effect of the model, and in using year instead to encapsulate all of that variability. Please include this explanation in the text, probably in the last paragraph of section 2.2.

Reply. We have included at the end of last paragraph of section 2.2. that: “Rather than using temperature statistics as predictor variables for each island through time, we preferred to include ꞌyearꞌ, which encapsulates any type of oceanographic variation through time”, as detailed in the next section.

L 240: I understand the concerns of the authors regarding simplicity of the figure. However, It is important that these plots speak in the same language as the figures and the models. As it stands right now, figure 2 is not particularly useful (beyond providing an outline of the islands) because it does not show effort. Furthermore, it does not show where the zeroes are. For example, as someone who has never been to El Hierro, I do not know if the north end of the island has no burrfish because there are none there, or because there were no dive surveys there. If the authors wish to maintain this spatial intra-island scale data in these figures, the data need to reflect 1) the dives with no sightings and 2) the sightings data expressed as SPUE. One way to do this without adding much complexity would be to have the size of the circle correlate to the number of hours of diving performed at a site, and a color gradient for SPUE. This would give the reader an idea of the spatial extent of the sampling effort as well as a true estimate of population gradients of the fish, which the authors discuss elsewhere to make some of their discussion points. Alternatively, the authors could eliminate the references to the intra-island spatial patterns in the manuscript and focus on the between-island patterns. This may be necessary as it appears in the supplementary data that there are no data on dives undertaken when fish were not sighted (so again, I cannot tell whether the spatial gaps in coverage were because of a lack of effort or because dives occurred but there were no fish sighted).

Reply. In this study, we are basically focusing on between-island trends. It’s true that intra-island gaps do exists in the observations, this may result from varying intensity in diving as a result of differences in habitat types. For example, large part of the continental shelfs around each island contain sandy areas where there is no diving. This is not a big issue, because our target species is a reef/rocky fish. We agree that Figure 2 is not very informative, and we can move it to the appendix if the reviewer insists. However, we still think this figure provides an overall picture on how sightings were distributed across the entire perimeter of each island.

L 298: The grammar of this first sentence needs to be fixed.

Reply. We have tried to clarify this (Lines 298-299).

L 316: If the authors wish to use Meridionalization I think it is important to describe it is a Mediterranean specific term (with the 2008 reference) for readers who are not from the Mediterranean region and may not be familiar with it. The authors’ response clarified the distinction and it would be good to clarify the text as well.

Reply. As the Reviewer points, we have included now that ‘meridionalization’ is specifically used to the Mediterranean Sea. Please, see line 311.

L342: “of” to “from”

Reply. “of” was changed to “from”

L348: replace “despite” with “though”

Reply. “despite” was replaced with “though”

L365: I disagree with the authors that the lack of accumulation of this fish around man-made structures or abundance gradients are evidence that it was not introduced by maritime traffic, as transport and aggregation are two separate processes. Given the long larval dispersal time of this species, is it not possible that untreated ballast water could play a role in larval (or even juvenile) dispersal? Furthermore, if the fish are not aggregating, doesn’t this make the point that they are able to freely disperse? I suggest removing this paragraph or including an analysis with “distance to port” included as a co-variate if the authors wish to make this strong of a case. Without data on effort across the island ( including where the fish are not seen) it is unclear how effort differs spatially. Perhaps the patterns really are that clear; but right now they don’t come through to a reader of the manuscript. I think revising figure 2 as requested above would be a good way to convince readers of the patterns the authors describe.

Reply. Despite we acknowledge the value of this idea, we do not agree. Since 2011, a large number of oil-platforms and drill vessels come to the industrial ports of the Canary Islands (i.e. Las Palmas Port, Santa Cruz Port, and Arinaga Port). Since this date, the University of Las Palmas and University of La Laguna have been carried out a surveillance program on marine fishes associated to these man-made structures. Many underwater visual census have been made within and around these ports to detect alien fish species. Importantly, no single of C. reticulatus has been detected to date, neither as juveniles nor as adults (see: Brito et al. 2011. Biological Invasions 13(10): 2185; Pajuelo et al. 2016. Journal of Marine Systems 163: 23-30; Triay-Portella et al. 2015. Cybium 39(3): 163-174; Falcón et al. 2015. Revista de la Academia Canaria de Ciencias 27(1): 67-82; Falcón et al. 2018. Revista de la Academia Canaria de Ciencias 30: 39-56).

L 372: Please also mention the lack of marine reserves in the text as you did in your response to the reviewer, since this is probably the main way that these fish would piggyback off of non-specific protections.

Reply. We have included that: “First, there is a lack of marine reserves targeting protection of this species.”

Supplementary material: I am a bit confused about the format of the dataset and would appreciate some clarification. Each row shows exactly one and only one burrfish sighted despite a wide range in effort.

Reply. Yes, each row corresponds to a sighting of the burrfish. This is denoted in the legend.

Were there never more than 1 burrfish sighted per operator per year, or per dive?

Reply. The burrfish is a rather solitary species. Hence, most sightings correspond to one individual. It is possible that the same individual might have been observed by the same diver at different times, or by a different operator. This is an error we cannot avoid.

Furthermore, what about locations where diving occurred but burrfish were never spotted?

Reply. It is plausible that certain rocky reefs are not ideal for the species. As we here demonstrate, most observations were from reefs with high relief structure (caves, ledges), so low relief reefs do not seem to provide an ideal habitat. This is, however, out of the scope of the paper.

Reviewer 3 Report

The changes look fine to me, I look forward to seeing this paper in print.
